# Machining Ti-6Al-4V Alloy Using Nano-Cutting Fluids: Investigation and Analysis

**Abdelkrem Eltaggaz** *[ID], **Ibrahim Nouzil and Ibrahim Deiab**

Advanced Manufacturing Laboratory, School of Engineering, University of Guelph,
Guelph, ON N1G2W1, Canada; inouzil@uoguelph.ca (I.N.); ideiab@uoguelph.ca (I.D.)
* Correspondence: aeltagga@uoguelph.ca

**Abstract:** Minimum Quantity Lubrication nanofluid (MQL-nanofluid) is a viable sustainable alternative to conventional flood cooling and provides very good cooling and lubrication in the machining of difficult to cut materials such as titanium and Inconel. The cutting action provides very difficult conditions for the coolant to access the cutting zone and the level of difficulty increases with higher cutting speeds. Furthermore, high compressive stresses, strain hardening and high chemical activity results in the formation of a 'seizure zone' at the tool-chip interface. In this work, the impact of MQL-nanofluid at the seizure zone and the corresponding effects on tool wear, surface finish, and power consumption during machining of Ti-6Al-4V was investigated. Aluminum Oxide ($Al_2O_3$) nanoparticles were selected to use as nano-additives at different weight fraction concentrations (0, 2, and 4 wt.%). It was observed that under pure MQL strategy there was significant material adhesion on the rake face of the tool while the adhesion was reduced in the presence of MQL-nanofluid at the tool-chip interface, thus indicating a reduction in the tool chip contact length (TCCL) and reduced seizure effect. Furthermore, the flank wear varied from 0.162 to 0.561 mm and the average surface roughness (Ra) varied from 0.512 to 2.81 μm. The results indicate that the nanoparticle concentration and the reduction in the seizure zone positively influence the tool life and quality of surface finish.

**Keywords:** MQL-nanofluid; seizure zone; analysis of variance; Ti-6Al-4V titanium alloy

## 1. Introduction

Many applications in the field of automotive and aeronautical industries utilize the difficult to cut materials such as titanium and Inconel due to their effective properties such as high corrosion resistance and better strength-to-weight ratio [1]. On the other hand, there are characteristics including low thermal conductivity and workpiece material hardness, which have negative effect on the machined surface roughness and tool life. An understanding of the influence of different machining design variables (e.g., cutting velocity, tool geometry, coolant strategy, feed rate, and depth of cut) has a significant impact in improving machinability, productivity, and reducing the total machining cost. Several studies have been conducted with an aim to specify the optimal machining variables and select a better coolant strategy as machining of difficult to cut materials is still facing different problems, particularly in the machining of ceramics, titanium alloys, polymers, nickel-based alloys, composite based materials and hardened steels [1–4]. The unique characteristics of titanium alloys (e.g., Ti-6Al-4V grade 5 and Ti-3Al 2.5 grade 12) such as, high thermal fatigue resistance, high erosion resistance and high melting temperature suit the purpose of many industrial applications [5]. However, the disposal of the generated heat during the machining is not suitable due to the low heat conductivity of these alloys [6,7]. The increase in machining generated heat over the critical limit leads to negative effects on the machined workpiece and the tool [8]. In general, improper tool wear behavior and surface integrity are correlated with the machining of titanium alloys [9]. The usage of cutting fluids during machining is mainly to take out the cutting generated heat.

The cutting fluids are extensively used in almost all kind of manufacturing industries. Apart from tool-workpiece interface cooling, cutting fluids also perform the lubrication regime and to wash away chips from the cutting zone during machining operation. Therefore, they account for generating good working conditions at the tool-workpiece interface. It also leads to the enhancement of several properties and process parameters like surface topology and cutting forces. Much research has been attempted to produce cutting fluids with superior tribological and thermal properties linked with the operator health and environment impacts. The past two decades have witnessed a collective research all over the globe for the development of a sustainable dry or near dry machining processes utilizing environmentally friendly coolants while eliminating the use of conventional coolants containing hazardous chemicals that affect the operator and the environment [10]. The simplicity, effectiveness and eco-friendly features of minimum quantity lubrication (MQL) have highlighted it as among the best possible alternatives to conventional cooling methods. MQL is currently utilized in many machining processes such as turning and grinding and the benefits of MQL to the machining performance of these processes is clearly indicated in the available literature [11]. The MQL approach is an environmentally friendly coolant strategy, but it has insufficient cooling capacity.

Alternatively, the use of minimum quantity lubrication (MQL) reduces the cutting fluid quantity during machining [9]. During machining difficult to cut materials, MQL is not a proper alternative of the flood coolant. To enhance the wettability and fluid thermal aspects, MQL-nano-fluid approach has been used [11,12]. Adding surfactant and Zeta potential for stability analysis is recommended to avoid the agglomeration of nanoparticles in such an approach. The aluminum oxide ($Al_2O_3$) gamma nano-additive has remarkable cooling and lubrication properties. The influence of nanofluids technology on the performance of metal cutting has been investigated in several studies. For instance, the $Al_2O_3$ nanoparticle has been used in micro-drilling of Inconel 718 [13]. MQL-nano-fluid with $Al_2O_3$ was also implemented during turning of austempered ductile iron (ADI) and Ti-6Al-4V [14,15]. It should be stated that MQL-nano-fluid showed promising results in enhancing the machining performance in all above-mentioned studies. Furthermore, multi-walled carbon nanotubes based nano-fluids with MQL showed better results in surface roughness and flank wear [16] compared to classical MQL and $Al_2O_3$ based nano-fluids during machining Ti-6Al-4V. In addition, many attempts have focused on employing the MQL-nanofluid technique when machining Inconel 718 [17,18], titanium [19] and austempered ductile iron (ADI) [20,21] and promising results have obtained in terms of tool wear, chip morphology, cutting power and surface quality.

The chip-tool interactions play a significant role in determining chip formation and in the rate and type of tool wear [22]. The cutting action provides very difficult conditions for the coolant to access the cutting edge and the level of difficulty increases with higher cutting speeds. Further, high compressive stresses, strain hardening and high chemical activity results in the formation of 'seizure zone' at the tool–chip interface [23]. Similar to solid phase welding or friction welding, simultaneous plastic deformation and high compressive stresses occurring while machining, provide the required conditions for the formation of metallurgical bonds between the workpiece (chip) and the tool. In the seizure zone, the tool and the workpiece essentially become a single piece of metal and the continued relative motion (chip formation) is caused by controlled fracture rather than by ordinary sliding friction. The characteristics of this controlled fracture mechanism, occurring as a result of seizure, determine the quality of surface and the 'integrity' of the material produced. Trent [24] conducted experimental investigations to study the mechanism of seizure and observed that, in the machining of steel with carbide tools, the workpiece material deposited on the tool, covering all the hills and valleys and in continues contact up to a certain distance from the cutting edge. Wallace and Boothroyd [25] conducted studies on aluminum alloy and observed two friction mechanisms at the interface: one as sliding friction and other as sticking friction (Seizure). In addition, Bailey [26] has also quoted references in support of the formation of sticking or 'seizure' between the tool and chip.

The inaccessibility of the coolant to the cutting edge is one of the reasons for the development of the seizure zone [23]. This is also supported by the study conducted by Hitomi et al. [27] where the authors state that adhesion at the tool flank can be reduced by implementing lubrication in the machining of cast iron. Despite many studies that have investigated the effect of MQL-nanofluid in the metal cut, limited articles in the open literature have deeply discussed the effect of nanoparticle concentrations on the machining performance of Ti-6Al-4V titanium alloy and seizure zone. Thus, this study is intended to cover this gap by investigating the effect of nanoparticle concentrations on the seizure zone and consequently on surface finish, tool life, and power consumption when machining Ti-6Al-4V. Furthermore, a clear tribological mechanism is presented and discussed to physically justify the nanoparticle concentration effects on the seizure zone and machining performance.

## 2. Materials and Methods

The experiments were conducted using vegetable oil as the MQL base fluid with $Al_2O_3$ as the dispersed nanoparticles in the machining of Ti-6Al-4V. The experimental setup is illustrated in Figure 1. Tables 1 and 2 detail the chemical composition and the mechanical properties of Ti-6Al-4V. A 0.5 mm depth of cut was maintained for all the machining trials over a length of 50 mm. For the MQL system, an Eco-Lubric booster system supplied an air-oil mist with 0.5 MPa air pressure and 40 mL/h nominal oil flow rate as employed in previous studies [20,21]. The properties of the MQL oil are as shown in Table 3.

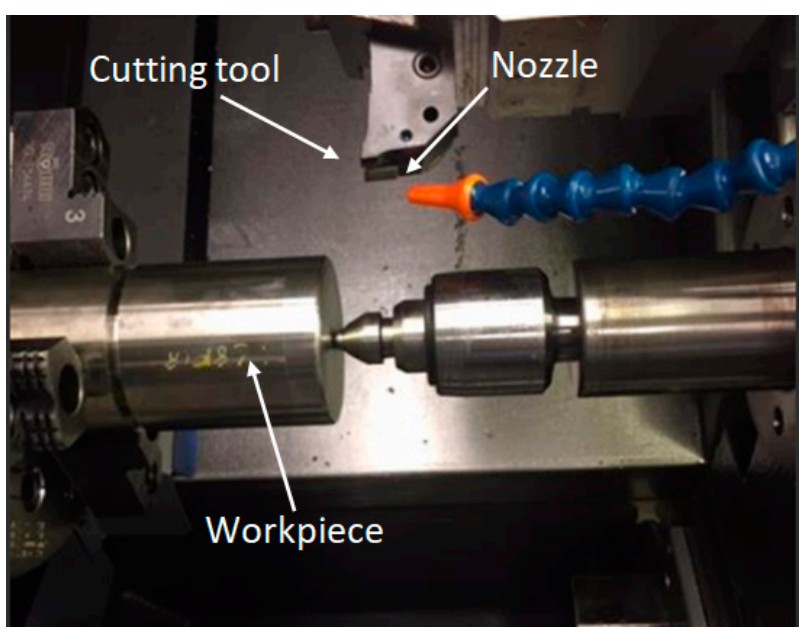

**Figure 1.** The experimental setup in detail.

**Table 1.** Chemical composition of Ti-6Al-4V.

| Alloy | Al | V | Fe | C | Ti |
|---|---|---|---|---|---|
| Ti-6Al-4V | 6.40% | 0.1% | 0.16% | 0.002% | 93.338% |

**Table 2.** Mechanical properties of Ti-6Al-4V titanium alloy.

| Ultimate Tensile Strength | Yield Tensile Strength | Elongation at Break | Modulus of Elasticity | Poisson's Ratio | Density |
|---|---|---|---|---|---|
| 950 MPa | 880 MPa | 14% | 113.8 GPa | 0.342 | 4.43 g/cm$^3$ |

**Table 3.** MQL oil properties.

| Properties | Description |
| --- | --- |
| Chemical description | Pure vegetable-based lubricant without any chemical modification |
| Health Risk | Environmentally friendly; no risk to operator health |
| Flash Point | 325 °C |
| Ignition Point | 365 °C |
| Density | 0.92 g/cm$^3$ |
| Viscosity | 70 cP (at 20 °C) |
| Partition Coefficient | <3% |

In this study, the Al$_2$O$_3$ nanoparticles with a 22 nm average diameter, 134 m$^2$/g specific surface areas, 92% purity was used as nano-additives due to its great tribological property and anti-toxic aspect. The soncators device was used in the dispersion of nanoparticles and the resulting nanofluid as shown in Figure 2.

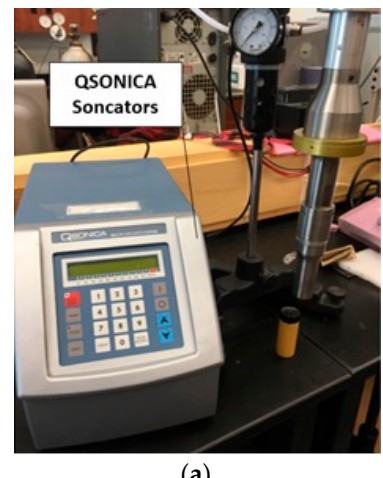 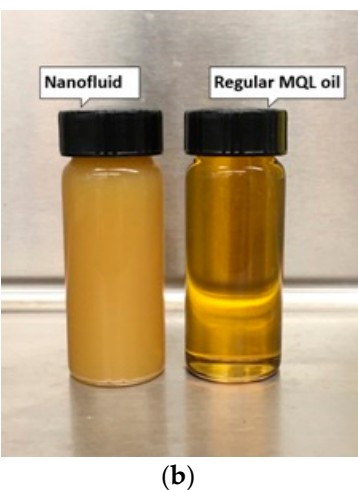

(**a**)                    (**b**)

**Figure 2.** (**a**) AQUASONIC-50HT device used to achieve the dispersion; (**b**) a regular MQL and MQL-nanofluid [19].

However, the dispersion of nanoparticles into the oil base fluid is challenging due to microscopic forces that are applied on nanoparticles. These forces such as gravity, Van der Waals dispersion forces, and density difference promote sedimentation and nanoparticle agglomeration [21]. The dispersion of nanoparticles into the base oil is crucial because it affects the viscosity and thermal conductivity of the resultant nanofluid-MQL. In this regard, to promote a sufficient dispersion of nanoparticles, it is recommended to use chemical or physical treatment such as surfactants [21] and in this study Sodium Dodecyl Sulfate (SDS) was utilized as a surfactant (0.2 gm). Surfactants are believed to make the nanoparticle performance more hydrophilic and to increase the surface charges of the nanoparticles, thereby increasing the repulsive forces between the nanoparticles. In order to evaluate the agglomerate of nanoparticles into the MQL-nanofluid, Zeta potential for stability analysis was also conducted. The Zetasizer nano-device was employed to estimate the zeta potential absolute values for nanofluids with two Al$_2$O$_3$ concentrations. The larger the value of the Zeta potential absolute, the better the dispersion of Al$_2$O$_3$ nanoparticles and a modeled level of zeta potential has been noticed. Flank wear measurements and surface roughness were obtained after each machining pass, using a Keyence VHX-5000 and a surface roughness tester, respectively. Surface roughness measurements were repeated three times for each surface and the average value was used. Scanning electron microscope (SEM), equipped with an Energy-Dispersive X-ray Spectroscopy (EDS) detector was used for high resolution imaging of the worn tools on both flank and rake faces. A power sight

manager device was utilized throughout the cutting operation to assist in recording the power consumption. Figure 3 presents the flow chart of the experimental setup.

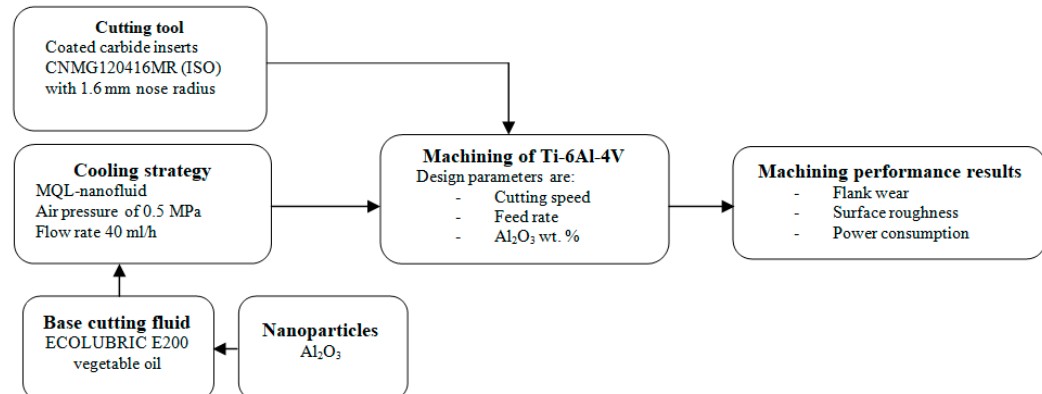

**Figure 3.** The flow chart of the experimental setup.

The cutting fluid usage when applied as a standard flood coolant can be an environmental concern; however, when employing the MQL approach, a certain amount of base oil was utilized producing very fine mist and significantly reducing the volume of coolant used. Safety measures recommended when using nanoparticles were followed to eliminate any health and environment concerns associated with utilizing nano-additives. In terms of the disposal process, the nano-fluids were carefully filtered after use and hazardous waste disposal steps were followed, as recommended by the Environmental Health and Safety (EHS) department [20,21]. Machining tests using an MQL nanofluid with different concentrations of volume fraction (0, 2, and 4 wt.%), various cutting speeds (120, 170, and 220 m/min) and feed rates (0.1, 0.15, and 0.2 mm/rev) were designed for the study. Cutting speed, feed rate and nanofluid concentration were selected as design variables for machining the Ti-6Al-4V titanium alloy. Each test was replicated three times and average values were obtained for more reliable data. In this study, three design parameters were selected with three levels each.

Table 4 illustrates the independent process factors studied and the corresponding levels assigned for the machining of titanium alloy workpiece. The orthogonal array L9 Taguchi method (L9OA) was applied during the cutting experiments. L9OA has 9 rows corresponding to the number of trials with 4 columns at 3 levels. The full factorial array (L27OA) was not applied to save time and cost. Experiments and the analysis of the variances (ANOVA) were conducted for each of the design parameters and labeled as A (cutting speed), B (feed rate), and C (nanoparticle concentration), as seen in Table 4.

**Table 4.** L9OA with respect to the studied design variables.

| Experiment No. | A: Cutting Speed (m/min) | B: Feed Rate (mm/rev) | C: Nanoparticle (wt.%) |
|---|---|---|---|
| 1 | 120 | 0.1 | 0 |
| 2 | 120 | 0.15 | 2 |
| 3 | 120 | 0.2 | 4 |
| 4 | 170 | 0.1 | 2 |
| 5 | 170 | 0.15 | 4 |
| 6 | 170 | 0.2 | 0 |
| 7 | 220 | 0.1 | 4 |
| 8 | 220 | 0.15 | 0 |
| 9 | 220 | 0.2 | 2 |

MINTAB 17 software was used to perform the ANOVA tests, which were used to explore the influences of design factors on tool wear, machined surface quality and power

consumption. Using the ANOVA, the importance of cutting parameters and nanoparticle concentrations was investigated with respect to the studied responses to discover the optimum combination of cutting parameters. All ANOVA analyses were conducted with a confidence level of 95% ($\alpha = 0.5$).

## 3. Results and Discussion

### 3.1. Seizure Zone in MQL-Nanofluid

The positive impact of lubrication on seizure effect and in the reduction in tool-chip contact length (TCCL) is well established in the literature [28]. However, the effect of nano MQL on seizure zone formation is not thoroughly realized. The sticking friction zone at the tool chip interface is known as seizure and the chip formation results from the material movement in the slipping zone that surrounds the seizure zone.

Under conditions of seizure, tool and workpiece materials are in intrinsic contact with no possible lubrication penetration at this zone, i.e., the workpiece material and tool essentially become one piece of metal. However, according to Trent [23], the cutting lubricant cans penetrates the slipping zone, where the apparent contact area is 1/1000 of the real contact area. Further, the seizure condition increases the contact temperature at the interface resulting in softer workpiece material and therefore increasing the tool-chip contact length (TCCL). The increased TCCL negatively impacts the tool life and machined surface quality. The experimental observations indicated that the nanofluid-MQL method showed reduced TCCL compared to the pure MQL as showed in Figure 4. It can be attributed to the improved the degree of sliding and reduced the friction force between the tool and machined surface.Two main mechanisms are initiated by the application of nanofluid droplets at the tool chip interface, i.e., rolling and ploughing mechanisms as shown in Figure 5.

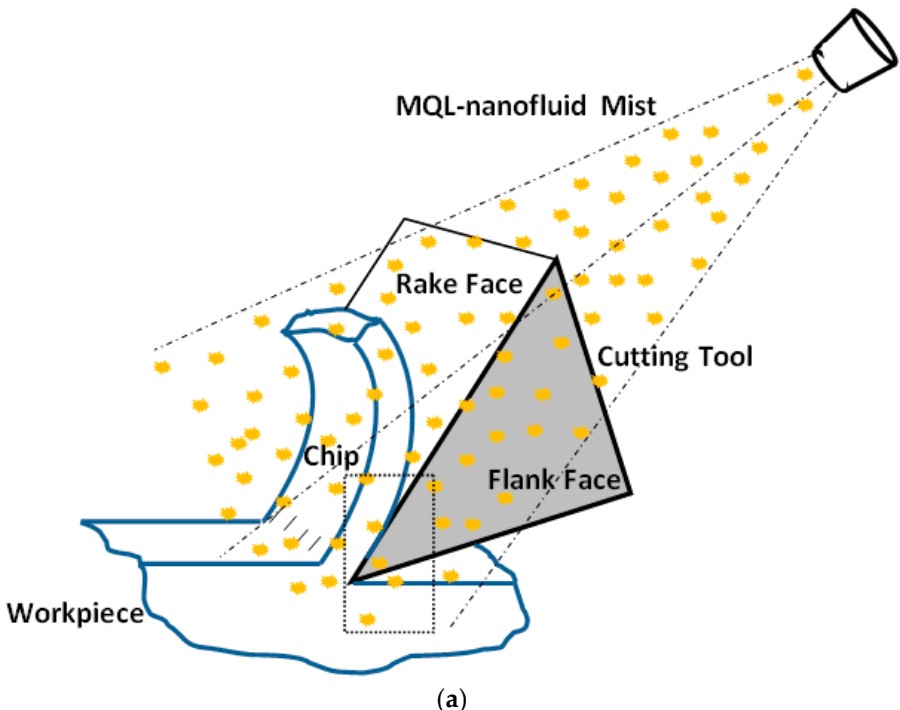

(a)

**Figure 4.** *Cont*.

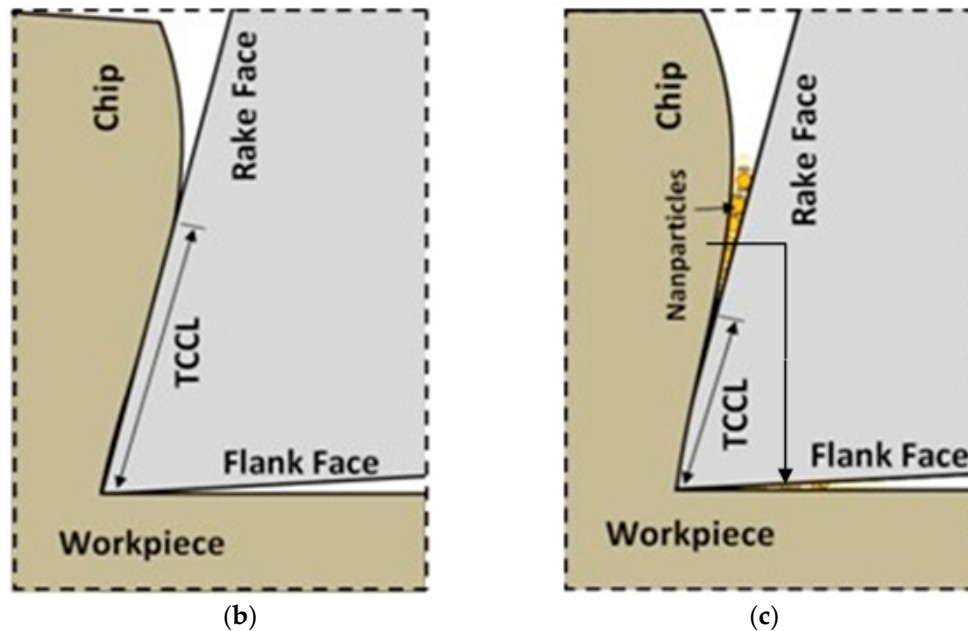

**Figure 4.** (**a**) 3D drawing with the actual MQL-nanofluid nozzle orientation during machining process (**b**) The TCCL at the tool-chip contacts with pure MQL (**c**) the TCCL with MQL-nanofluid.

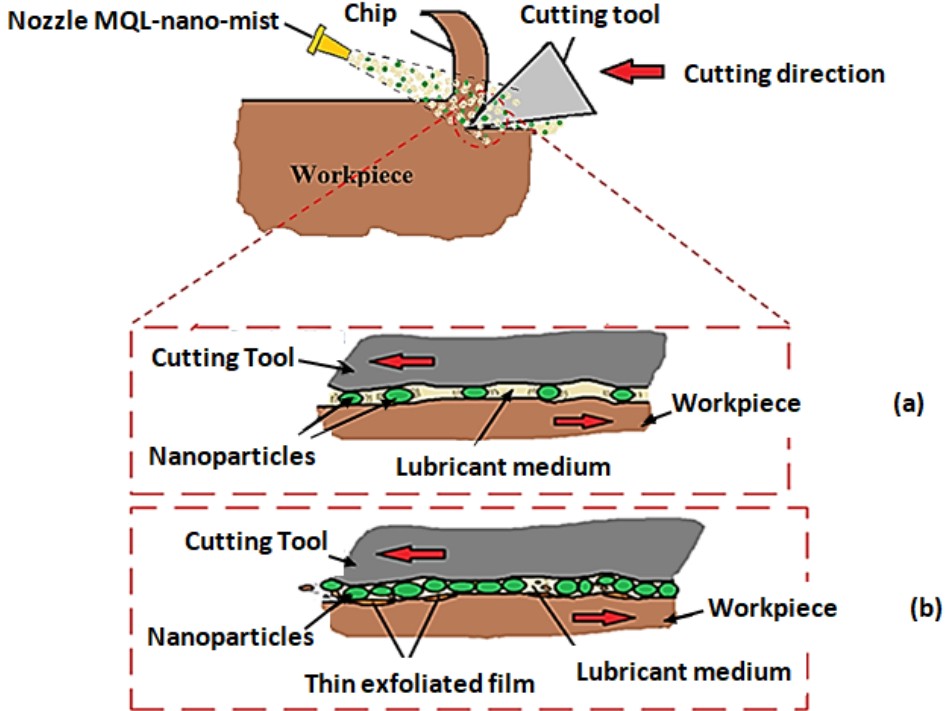

**Figure 5.** The MQL-nanofluids mechanisms (**a**) rolling effect and (**b**) ploughing effect.

These mechanisms are related to the nanoparticle concentration. The addition of spherical nanoparticles turns the sliding friction at the interface into a combination of sliding and rolling friction [21]. This rolling action results in less contact between the workpiece and cutting tool, thereby shortening the TCCL. However, increasing the nanoparticles in the cutting zone may limit the rolling mechanism due to the high compression that results with an increase in the nanoparticle concentration. This induces a different mechanism known as the ploughing mechanism. In the ploughing mechanism, the nanoparticles are pushed away from the tool-workpiece zone by other particles atomized through the MQL nozzle.

The ploughing action reduces the lubricity of the nanofluid jet and increases the coefficient of friction at the interface [29]. The nanoparticle concentration is therefore considered a critical design variable to be optimized when using MQL nanofluid technology, as controlling this variable would have a significant influence on the induced mechanisms (i.e., rolling or ploughing). Figures 6–8 show the pictures of tool inserts taken at the end of machining trails (cutting length of 200 m). Scanning electron microscope (SEM) and Energy Dispersive X-ray Spectroscopy (EDS) images show tool wear and material adhesion on the rake face, respectively. Figure 6 shows the adhesion and abrasion wear on the tool when employing the regular MQL technique at a cutting speed of 220 m/min and a feed rate of 0.15 mm/rev, while Figures 7 and 8 represent the MQL nanofluid concentrations of 2% at a cutting speed of 220 m/min and a feed rate of 0.2 mm/rev and MQL nanofluid concentrations of 4% at a cutting speed of 220 m/min and a feed rate of 0.1 mm/rev, respectively.

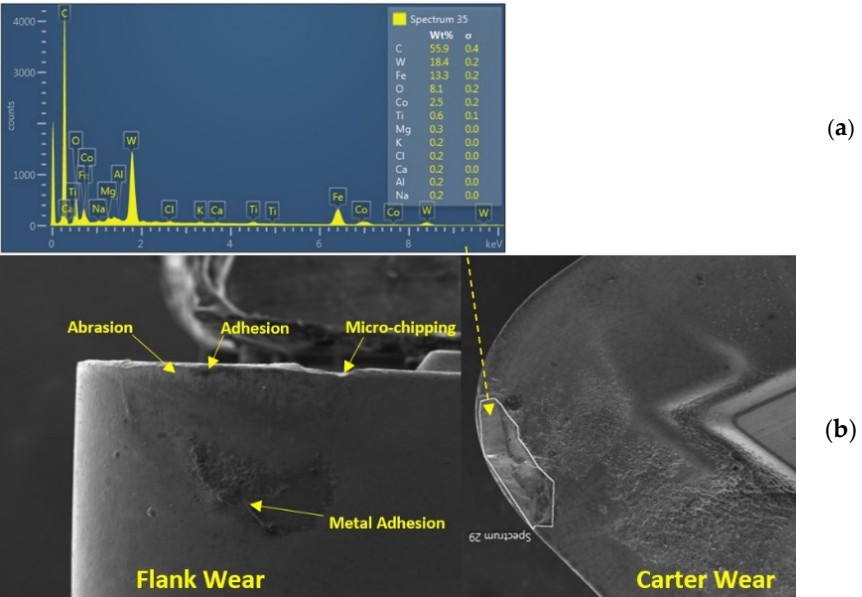

**Figure 6.** EDS (**a**) and SEM (**b**) pictures of the tool after using regular MQL at a cutting speed of 220 m/min and a feed rate of 0.15 mm/rev.

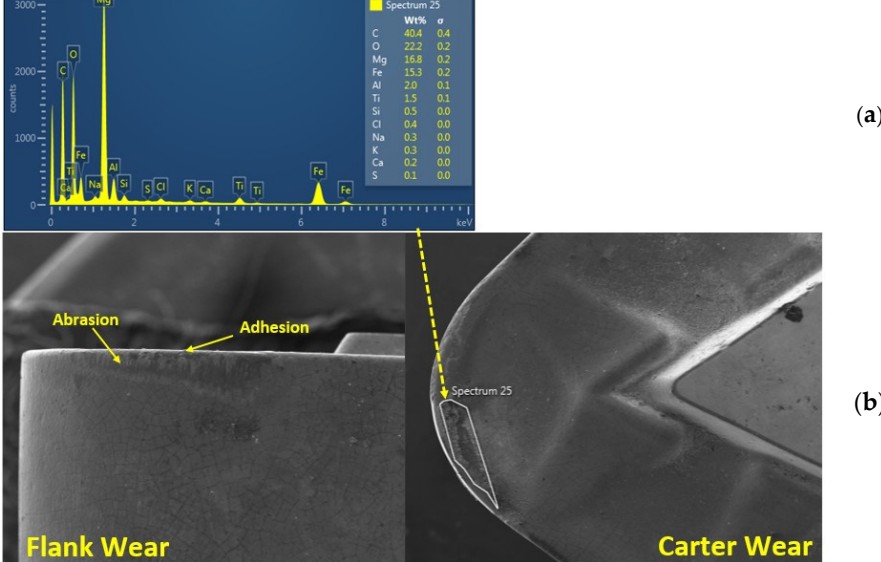

**Figure 7.** EDS (**a**) and SEM (**b**) pictures of the tool after using MQL-Nanofluid 2% at a cutting speed of 220 m/min and a feed rate of 0.2 mm/rev.

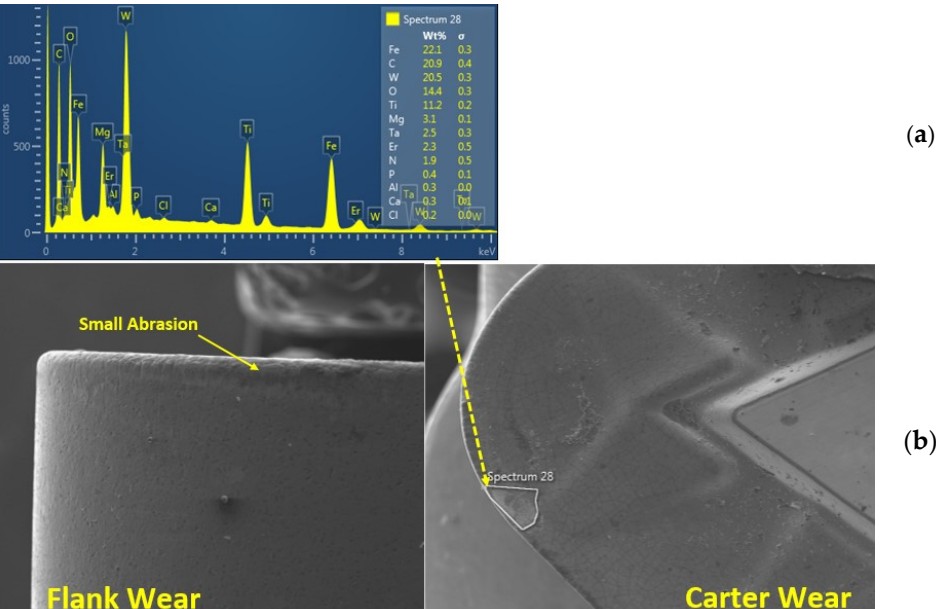

**Figure 8.** EDS (**a**) and SEM (**b**) pictures of the tool after using MQL-Nanofluid 4% at a cutting speed of 220 m/min and a feed rate of 0.1 mm/rev.

As seen in the EDS (a) and SEM (b) images, the material adhesion on the tool, attributed to the sticking friction zone (Seizure zone), is considerably reduced by the application of nanofluid MQL technique. The high temperatures in the cutting zone and the chemical reaction of workpiece materials results in higher TCCL and greater seizure effect while using regular MQL technique.

Figures 7 and 8 present the condition of the cutting tool after using MQL-nanofluid. It is observed that adhesion of material on the tool rake face is reduced due to the action of the nanoparticles, which indicates reduced TCCL and seizure effect. Furthermore, it is also noted that the crater wear was higher with the regular MQL (Figure 6) and reduced with the application of nanofluid (Figures 7 and 8). This confirms that the nanofluid MQL successfully reduces the TCCL and reduces the impact of the seizure zone in the machining of titanium.

### 3.2. Effect on Tool Wear, Surface Roughness and Power Consumption

The continues chip formation under seizure conditions is due to the process of controlled fracture and the character of this fracture mechanism determines the surface finish and integrity of the machining process. It is important to study the impact of MQL-nanofluid and seizure on the desired machining characteristics and Table 5 illustrates the results for the quality characteristics that were investigated.

In terms of tool wear, the lowest maximum flank wear ($VB_{max}$) was noticed at a speed of 120 m/min, a feed rate of 0.2 mm/rev, and a nanoparticle concentration of 4 wt.%. As per machining principle, speed significantly affected the tool wear due to the high heat generation in the cutting zone at increased speed. However, nanoparticle concentration also had a significant effect on the max flank wear. In Figure 9, it is observed that the increased nanoparticle concentration greatly reduced the tool wear. Furthermore, analysis of variance (Table 6) on the maximum flank wear also showed a 42.65% contribution of nanoparticles with regard to the tool wear.

**Table 5.** The machining parameters levels and the output measurements. Where: A represents cutting speed (m/min), B represents feed rate (mm/rev), and C represents nanoparticles concentration (wt.%).

| Exp. No | A | B | C | Surface Roughness Ra (µm) | Max Flank Wear $VB_{max}$ (mm) | Power Consumption P (W-h) |
|---|---|---|---|---|---|---|
| 1 | 1 | 1 | 1 | 0.891 | 0.236 | 2184 |
| 2 | 1 | 2 | 2 | 0.852 | 0.241 | 2141 |
| 3 | 1 | 3 | 3 | 1.692 | 0.162 | 1959 |
| 4 | 2 | 1 | 2 | 0.512 | 0.192 | 2079 |
| 5 | 2 | 2 | 3 | 1.421 | 0.171 | 2106 |
| 6 | 2 | 3 | 1 | 2.812 | 0.271 | 1989 |
| 7 | 3 | 1 | 3 | 0.563 | 0.211 | 2084 |
| 8 | 3 | 2 | 1 | 1.890 | 0.561 | 2248 |
| 9 | 3 | 3 | 2 | 0.951 | 0.242 | 2020 |

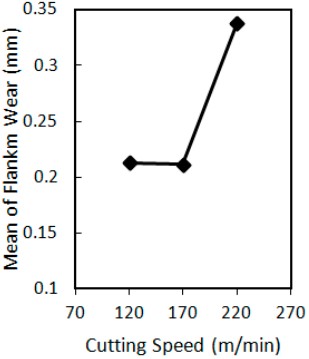 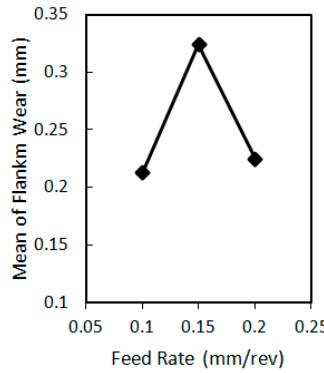 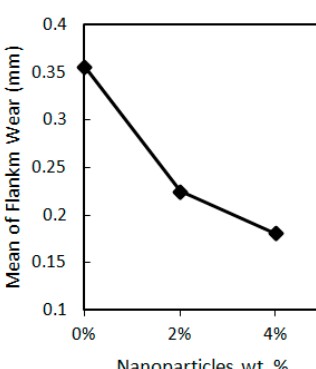

**Figure 9.** The main effects plot for maximum flank wear.

**Table 6.** Analyses of variance for maximum flank wear ($VB_{max}$).

| Sources | Degrees of Freedom | Sum of Squares | Mean of Squares | Contribution Percentage % |
|---|---|---|---|---|
| Cutting Speed | 2 | 0.031672 | 0.015836 | 27.25 |
| Feed | 2 | 0.022406 | 0.011203 | 19.27 |
| Nanoparticle wt.% | 2 | 0.049576 | 0.024788 | 42.65 |
| Error | 2 | 0.012566 | 0.006283 | 10.81 |
| Total | 8 | 0.116221 | | 100 |

This effect on tool wear was likely due to the reduced seizure effect, good thermal conductivity of the nanoparticle and its ability to penetrate the chip/workpiece zone. The improvement in tool wear could also be affected by the rolling action of the nanoparticles, which changed the wear behavior from sliding friction to rolling friction.

The lowest surface roughness was noted at a feed rate of 0.1 mm/rev, a cutting speed of 170 m/min, and a nanoparticle concentration of 2 wt.%. Feed rate and nanoparticles wt.% were found to have the greatest impact on surface roughness quality (see Figure 10), contributing 46.55% and 40.75%, respectively (see Table 7). It is known that the feed rate has a significant effect on the surface roughness in machining processes.

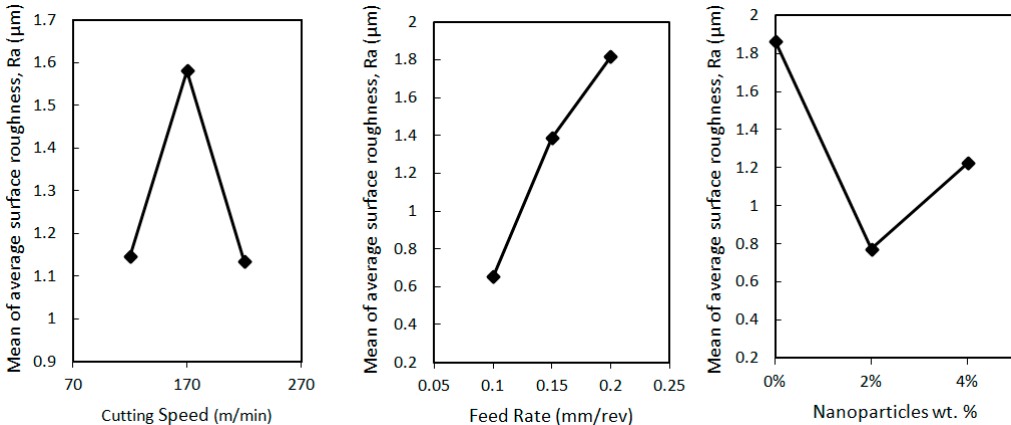

**Figure 10.** The main effects plot for surface roughness (Ra).

**Table 7.** Analysis of variance for surface roughness (Ra).

| Sources | Degrees of Freedom | Sum of Squares | Mean of Squares | Contribution Percentage % |
|---|---|---|---|---|
| Cutting Speed | 2 | 0.0373 | 0.186 | 8.45 |
| Feed | 2 | 2.0551 | 1.027 | 46.55 |
| Nanoparticle wt.% | 2 | 1.799 | 0.899 | 40.75 |
| Error | 2 | 0.187 | 0.093 | 4.25 |
| Total | 8 | 4.414 | | 100 |

In addition, the impact of nanoparticle concentration can be attributed to the ability of nanoparticles to penetrate the chip and workpiece surface and settle in any micro-grooves or slits. This is known as the surface protective effect of nano-mist. The increase in nanoparticle concentration to 4 wt.% caused a reduction in the surface roughness and this is attributed to the ploughing effect of excessive nanoparticles. A similar result of lower surface roughness at increased nanoparticle concentration is reported in literature [21]. Therefore, selecting the appropriate nanoparticle concentration would help to achieve better frictional behavior and surface finish, while avoiding the drastic influence of the ploughing mechanism.

Regarding the power consumption analysis (see Table 7), the lowest power consumption was obtained at a feed rate of 0.2 mm/rev, a cutting speed of 170 m/min, and a nanoparticle concentration of 4 wt.%. Feed rate and nanoparticle wt.% were found to have the highest impact on the measured power consumption; however, the speed still had an acceptable statistical contribution of about 7.8%, as can be seen in Figure 11 and Table 8.

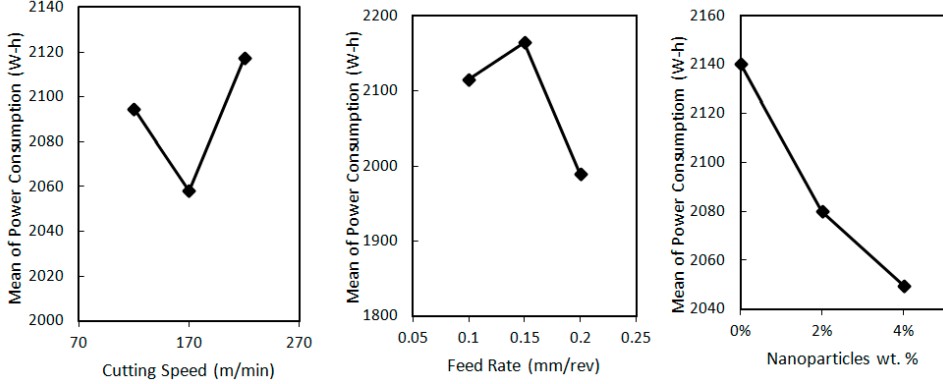

**Figure 11.** The main effects plot for power consumption.

**Table 8.** Analysis of variance for power consumption (P).

| Sources | Degrees of Freedom | Sum of Squares | Mean of Squares | Contribution Percentage % |
|---|---|---|---|---|
| Cutting Speed | 2 | 5378.667 | 2689.33 | 7.8 |
| Feed | 2 | 49,252.67 | 24,626.33 | 71.3 |
| Nanoparticle wt.% | 2 | 12,780.67 | 6390.33 | 18.5 |
| Error | 2 | 1664 | 832 | 2.4 |
| Total | 8 | 69,076 | | 100 |

It was noted that increasing the nanoparticle concentrations played a vital role in decreasing the induced friction as the nanoparticles act as spacers at the tool-workpiece contact, reduce the TCCL and therefore significantly affect the measured power consumption. However, Ghaednia et al. [29] reported that the increase in nanoparticles concentration would increase the particles abrasive induced wear. Further, as observed in the analysis of surface finish, higher concentration of nanoparticles caused a reduction in the surface quality and this is primarily due to the abrasive wear induced by the nanoparticles.

Ultimately, the MQL-nanofluid approach is an effective interactive technique for lubricating and cooling of a machining process and can be used as a sustainable cooling strategy during machining Ti-6Al-4V titanium alloy.

## 4. Conclusions

In this work, MQL nanofluid technology was employed when machining Ti-6Al-4V. Cutting speed, feed rate, and different nanoparticle concentrations (wt.%) was selected as the design variables and their impact on seizure zone was investigated. Furthermore, three machining outputs were investigated in this study: machined surface quality, tool life, and power consumption. The following conclusions are made from the study:

- It was observed that employing the MQL-nanofluid strategy reduced the seizure zone. The material adhesion on the rake face of the tool was found to be considerably less on application of nanofluid on the tool chip interface when compared to pure MQL.
- ANOVA tests were conducted to analyze the design variable effects on tool wear, surface quality and power consumption. Accordingly, the feed rate and the nanoparticle concentration were the most significant parameters in determining the surface roughness at contribution rates of 46.55% and 40.75%, respectively. The ANOVA analysis also showed that the nanofluid at 2 and 4 wt.% concentration of $Al_2O_3$ provided better results than the regular MQL in determining surface roughness. However, nanofluid at 2 wt.% performed much better than 4 wt.% and this attributed to the ploughing effect and abrasive action of the nanoparticles.
- Tool wear was significantly reduced by the application of nanoparticles. ANOVA analysis indicated that the nanoparticle concentration influenced the flank wear at a contribution rate of 42.65%. The reduced seizure zone, rolling friction and reduced interface temperature due to high thermal conductivity of the nanoparticles influenced the increased tool life.
- In terms of the power consumption, the lowest power consumption was noticed at a feed rate of 0.2 mm/rev, a cutting speed of 170 m/min, and nanoparticle concentration of 4 wt.%. The high particle concentrations resulted in a very low coefficient of friction at the interface, resulting in low contact forces and consequently lower power consumption.

MQL-nanofluid showed better results in terms of improving the tool performance, surface quality, and power consumption when compared to the pure MQL. In the future, more work could be done to determine the critical nanoparticle concentration needed to achieve maximum frictional enhancement while avoiding the drastic influence of the ploughing mechanism.

**Author Contributions:** Introduction, A.E.; methodology, A.E.; investigation, A.E.; data curation, A.E.; writing—original draft preparation, A.E.; writing—review and editing, I.D.; visualization, I.N.; supervision, I.D.; funding acquisition, I.D. All authors have read and agreed to the published version of the manuscript.

**Funding:** This research was funded by Natural Sciences and Engineering Research Council of Canada (NSERC) grant number 401363.

**Institutional Review Board Statement:** Not applicable.

**Informed Consent Statement:** Not applicable.

**Data Availability Statement:** Not applicable.

**Conflicts of Interest:** The authors declare no conflict of interest.

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
