# Peer review of "Machining Ti-6Al-4V Alloy Using Nano-Cutting Fluids: Investigation and Analysis"

_jmmp, doi:10.3390/jmmp5020042_

Round 1
Reviewer 1 Report
Abstract: MQL-nanofluid - clarification of the designation needed
surface roughness – which parameter? (Ra, I guess)
Line 27: the difficult-to-cut materials – give some specific examples
Line 29: high hardness, which have negative effect on the machined surface roughness - this is not always true, the carbide turning tool inserts have high hardness, yet they allow obtaining a roughness comparable to that obtained in the grinding process
Lines 37-38: titanium alloys – which alloys?
Line 47: what about the damping of vibrations in the tool-workpiece interface?
Line 51-53: these fluids have some detrimental effects on both human healths as well as on the environment due to the presence of toxic chemicals - it depends on the type and composition of the coolant. Currently, cooling liquids based on biodegradable plant-derived ingredients are being developed
Line 71: MQL-nano-fluid approach - what about the agglomeration of nanoparticles in such a nano-fluid?
Figure 1 - does the figure position result from correct formatting?
Figure 2. This is a figure. Schemes follow the same formatting. -??? Figure 2. The flow chart of the experimental setup. (should be - I guess)
Table 1: there is: (wt%) – should be: (wt.%)
Line 199: there is: Figure 2. TCCL at the tool-chip contact (a) with pure MQL (b) with MQL-nanofluid – should be: Figure 3. TCCL at the tool-chip contact (a) with pure MQL (b) with MQL-nanofluid
Fig. 3b) – Are there nanoparticles below the Flank Face?
Figure 3 at the bottom is something misleading: as if the chip was permeable to the nano-coolant - maybe an additional drawing in a 3D version with the actual orientation of the coolant flow relative to the contact zone knife - workpiece - the resulting chip, would be useful here to better illustrate the real situation
Figures 5, 6, 7 – it is needed to enter the designation a and b for the drawings in the components and in the description, specify which is SEM and which EDS, a layman will not be able to resolve it ...
Table 2 - does the table position result from correct formatting? in the description under the table (add, for example, a superscript to the markings in the table), it is necessary to explain what A, B, C means
Figure 8, 9, 10 - component figures must be horizontally separated from each other and the description of the horizontal axes must be under these axes, otherwise it is illegible and misleading. What do the dashed lines in the figures mean - it would have to be explained in the captions of the drawings
Conclusion
there is no assessment of the impact of the measures taken to prevent agglomeration of nanoparticles in the coolant on the tested surface parameters of the processed elements
Author Response
Author's Reply to the Review Report (Reviewer 1)
General response
The authors sincerely thank the reviewer for constructive suggestions and valuable comments, which were of great help in revising the manuscript. Accordingly, the revised manuscript has been improved with new information and additional interpretations. The all needful changes are made.
Comments:
- Abstract: MQL-nanofluid - clarification of the designation needed
Authors’ reply: The authors would like to thank the reviewer for his comment. Please see the revised manuscript, where the required changes have been done.
- Surface roughness – which parameter? (Ra, I guess)
Authors’ reply: The authors would like to thank the reviewer for his comment. Please see the revised manuscript, Ra has been added after surface roughness.
- Line 27: the difficult-to-cut materials – give some specific examples
Authors’ reply: The authors would like to thank the reviewer for his comment. Examples of difficult-to-cut materials have been given in the revised manuscript.
- Line 29: high hardness, which have negative effect on the machined surface roughness - this is not always true, the carbide turning tool inserts have high hardness, yet they allow obtaining a roughness comparable to that obtained in the grinding process
Authors’ reply: The authors would like to thank the reviewer for his comment. The high hardness is related to workpiece material not cutting tool. The statement has been clarified in the revised manuscript.
- Lines 37-38: titanium alloys – which alloys?
Authors’ reply: The authors would like to thank the reviewer. Titanium alloys have been stated in the revised manuscript.
- Line 47: what about the damping of vibrations in the tool-workpiece interface?
Authors’ reply: The authors would like to thank the reviewer. Dynamics of machining process has not been studied in the current manuscript.
- Line 51-53: these fluids have some detrimental effects on both human healths as well as on the environment due to the presence of toxic chemicals - it depends on the type and composition of the coolant. Currently, cooling liquids based on biodegradable plant-derived ingredients are being developed
Authors’ reply: The authors would like to thank the reviewer. Authors agree with reviewer and statement has been removed from the revised manuscript.
- Line 71: MQL-nano-fluid approach - what about the agglomeration of nanoparticles in such a nano-fluid?
Authors’ reply: The authors would like to thank the reviewer. Avoiding agglomeration and nanofluid stability has been discussed in section 2, please see the revised manuscript.
- Figure 1 - does the figure position result from correct formatting?
Authors’ reply: The authors would like to thank the reviewer. The position of Figure 1 has been corrected, please see the revised manuscript.
- Figure 2. This is a figure. Schemes follow the same formatting. -??? Figure 2. The flow chart of the experimental setup. (should be - I guess)
Authors’ reply: The authors would like to thank the reviewer. The caption under Figure 2 has been fixed, please see the revised manuscript.
- Table 1: there is: (wt%) – should be: (wt.%)
Authors’ reply: The authors would like to thank the reviewer. The comment has been addressed, please see the revised manuscript.
- Line 199: there is: Figure 2. TCCL at the tool-chip contact (a) with pure MQL (b) with MQL-nanofluid – should be: Figure 3. TCCL at the tool-chip contact (a) with pure MQL (b) with MQL-nanofluid
Authors’ reply: The authors would like to thank the reviewer. The comment has been addressed, please see the revised manuscript.
- 3b) – Are there nanoparticles below the Flank Face?
Authors’ reply: The authors would like to thank the reviewer. During the cooling process, a number of nanoparticles penetrated into workpiece/tool flank face contact.
- Figure 3 at the bottom is something misleading: as if the chip was permeable to the nano-coolant - maybe an additional drawing in a 3D version with the actual orientation of the coolant flow relative to the contact zone knife - workpiece - the resulting chip, would be useful here to better illustrate the real situation
Authors’ reply: The authors would like to thank the reviewer for his valuable suggestion. 3D version with actual orientation of the MQL-nanofluid coolant has been added to the Figure, please see the revised manuscript.
- Figures 5, 6, 7 – it is needed to enter the designation a and b for the drawings in the components and in the description, specify which is SEM and which EDS, a layman will not be able to resolve it ...
Authors’ reply: The authors would like to thank the reviewer for his comment. Please see the revised manuscript, where the needed changes have been done.
- Table 2 - does the table position result from correct formatting? in the description under the table (add, for example, a superscript to the markings in the table), it is necessary to explain what A, B, C means
Authors’ reply: The authors would like to thank the reviewer for his comment. Please see the revised manuscript, where the position of Table 2 has been corrected and A, B, and C have been explained under the table.
- Figure 8, 9, 10 - component figures must be horizontally separated from each other and the description of the horizontal axes must be under these axes, otherwise it is illegible and misleading. What do the dashed lines in the figures mean - it would have to be explained in the captions of the drawings
Authors’ reply: The authors would like to thank the reviewer. Please see the revised manuscript, where the all required have been changed.
- There is no assessment of the impact of the measures taken to prevent agglomeration of nanoparticles in the coolant on the tested surface parameters of the processed elements
Authors’ reply: The authors would like to thank the reviewer. Agglomeration assessment has been tested by the Zeta potential analysis; please see the revised manuscript.
Reviewer 2 Report
The manuscript presents a study in the use of nanoparticles in MQL lubrication. the manuscript is well structured however a series of improvements must be done on the manuscript prior to acceptance.
The machine tool used as well as all the relevant equipment and experimental setup must be included in the manuscript.
The depth of cut as well as the specification of the material must be stated.
The DOE set up must be revised. at its current state it is not complete. An L9 orthogonal setup includes 12 tests rather than the 9 presented. This allows for the calculation for the uncertainty in the measurements and it is crucial for the validity of the results. The reviewer would suggest using a full factorial design as it allows for pairwise comparisons between different concentrations of nanoparticles.
The repeats of the surface roughness measurement tests must be stated.
the contents of figures 5-7 must be presented with the rest of the cutting conditions as they are not compairable without them.
The length of cut or time in cut must be stated for each trial.
Author Response
Author's Reply to the Review Report (Reviewer 2)
General response
The authors sincerely thank the reviewer for constructive suggestions and valuable comments, which were of great help in revising the manuscript. Accordingly, the revised manuscript has been improved with new information and additional interpretations. The all needful changes are made.
Comments:
- The machine tool used as well as all the relevant equipment and experimental setup must be included in the manuscript.
- Authors’ reply: The authors would like to thank the reviewer for his valuable comment. Figure of experimental setup has been added to the revised manuscript.
- The depth of cut as well as the specification of the material must be stated.
Authors’ reply: The authors would like to thank the reviewer. The depth of cut has been stated and the specification of the materials has been added to the revised manuscript.
- The DOE set up must be revised. at its current state it is not complete. An L9 orthogonal setup includes 12 tests rather than the 9 presented. This allows for the calculation for the uncertainty in the measurements and it is crucial for the validity of the results. The reviewer would suggest using a full factorial design as it allows for pairwise comparisons between different concentrations of nanoparticles.
Authors’ reply: The authors would like to thank the reviewer for his suggestion. The authors agree with the reviewer. However, due to the limited access to the laboratory more tests cannot be achieved. Further, the results gathered from the tests conducted are sufficient in establishing the results.
- The repeats of the surface roughness measurement tests must be stated.
Authors’ reply: The authors would like to thank the reviewer. The repeats of the surface roughness measurements have been stated in the revised manuscript.
- The contents of figures 5-7 must be presented with the rest of the cutting conditions as they are not comparable without them.
Authors’ reply: The authors would like to thank the reviewer for his comment. The cutting conditions have been presented for figures 5-7 based on the orthogonal array design.
- The length of cut or time in cut must be stated for each trial.
Authors’ reply: The authors would like to thank the reviewer. The length of cut has been stated in the revised manuscript.
Reviewer 3 Report
The article deals with an actual and interesting issue of improvement of milling process of titanium alloys by minimum quantity lubrication with nanoparticles. The used methodology is correct, and the results are clear.
However, there are some small errors:
- lines 6 & 7 are author affiliation number 2; however, all authors are referred to affiliation 1 only. Please, check the affiliations.
- line 119 - there is written Ti-6Al-4v instead of Ti-6Al-4V (lowercase vanadium).
- line 150 - please, check the name of Figure 2. The name "This is a figure. Schemes follow the same formatting." obviously is not correct.
- line 199 - it is Figure 3 (not Figure 2).
- line 237 - there is a reference to Figure 8, but obviously, it should be Figure 5.
- chapter 3.1 - there was not mentioned the cutting conditions of the observed cutting tools (cutting speed, feed rate, machining time). I recommend adding them.
- Table 2 - I recommend specifying which surface roughness parameter was measured (probably Ra).
- line 292 - it is Table 5 (not Table 4).
- line 304 - it is Figure 11 (not Figure 10).
- please unify the terminology:
#1 - line 9: "difficult to cut", lines 27, 69: "difficult-to-cut", line 35: "hard-to-cut";
#2 - aluminum oxide - lines 15, 119, 123, 139, 140, 323: "Al2O3", lines 71, 74, 80: "Al2O3";
#3 - MQL - line 63: "lubrication", line 68: "lubricant";
#4 - Energy-Dispersive X-ray Spectroscopy - line 144: "EDX", lines 219, 222, 225, 231, 233: "EDS";
#5 - weight percentages - line 160, 266, 267, 273, 288, 310, 322, 324, 333: "wt. %" (dot & space), Table 1: "wt% (no dot & no space)", lines 224, 225, 231, 233: "%", line 250: "wt.%" (dot & no space);
#6 - in all work was used "workpiece material" (e.g. lines 99, 192), but in lines 187, 188, 228 was used "work material";
#7 - singular/plural - line 217: "Figure 5-7", line 224: "figure 6 and 7", line 234: "Figures 6 and 7";
#8 - in all work was used "nanoparticle" (e.g. line 254), but in line 252 was used "nano particle" (with space).
Author Response
Author's Reply to the Review Report (Reviewer 3)
General response
Authors sincerely thank the reviewer for constructive suggestions and valuable comments, which were of great help in revising the manuscript. Accordingly, the revised manuscript has been improved with new information and additional interpretations. The all needful changes are made.
Comments:
- Lines 6 & 7 are author affiliation number 2; however, all authors are referred to affiliation 1 only. Please, check the affiliations.
Authors’ reply: The authors would like to thank the reviewer. Please accept our apologize for the misunderstanding. The typo error has been removed in the revised manuscript.
.
- Line 119 - there is written Ti-6Al-4v instead of Ti-6Al-4V (lowercase vanadium).
Authors’ reply: The authors would like to thank the reviewer. The typo error has been fixed in the revised manuscript.
- Line 150 - please, check the name of Figure 2. The name "This is a figure. Schemes follow the same formatting." obviously is not correct.
Authors’ reply: The authors would like to thank the reviewer. The needful correction has been made in the revised manuscript.
- Line 199 - it is Figure 3 (not Figure 2).
Authors’ reply: The authors would like to thank the reviewer. The Figure number has been corrected in the revised manuscript.
- Line 237 - there is a reference to Figure 8, but obviously, it should be Figure 5.
Authors’ reply: The authors would like to thank the reviewer. The Figure number has been corrected in the revised manuscript.
- Chapter 3.1 - there was not mentioned the cutting conditions of the observed cutting tools (cutting speed, feed rate, machining time). I recommend adding them.
Authors’ reply: The authors would like to thank the reviewer for his/her recommendation. The cutting parameters and cutting length have been stated in the revised manuscript.
- Table 2 - I recommend specifying which surface roughness parameter was measured (probably Ra).
Authors’ reply: The authors would like to thank the reviewer for his/her comment. The surface roughness has been specified in the revised manuscript.
- Line 292 - it is Table 5 (not Table 4).
Authors’ reply: The authors would like to thank the reviewer for his/her comment. The Table number has been changed in the revised manuscript.
- Line 304 - it is Figure 11 (not Figure 10).
Authors’ reply: The authors would like to thank the reviewer for his/her comment. The Figure number has been changed in the revised manuscript.
- Please unify the terminology:
#1 - line 9: "difficult to cut", lines 27, 69: "difficult-to-cut", line 35: "hard-to-cut";
#2 - aluminum oxide - lines 15, 119, 123, 139, 140, 323: "Al2O3", lines 71, 74, 80: "Al2O3";
#3 - MQL - line 63: "lubrication", line 68: "lubricant";
#4 - Energy-Dispersive X-ray Spectroscopy - line 144: "EDX", lines 219, 222, 225, 231, 233: "EDS";
#5 - weight percentages - line 160, 266, 267, 273, 288, 310, 322, 324, 333: "wt. %" (dot & space), Table 1: "wt% (no dot & no space)", lines 224, 225, 231, 233: "%", line 250: "wt.%" (dot & no space);
#6 - in all work was used "workpiece material" (e.g. lines 99, 192), but in lines 187, 188, 228 was used "work material";
#7 - singular/plural - line 217: "Figure 5-7", line 224: "figure 6 and 7", line 234: "Figures 6 and 7";
#8 - in all work was used "nanoparticle" (e.g. line 254), but in line 252 was used "nano particle" (with space).
Authors’ reply: The authors have carefully addressed all the above comments. Thanks for giving us the opportunity for the revision.
Round 2
Reviewer 2 Report
The MQL oil specification must be also included for the cross-validation of the results.
The reason behind the change in the values on table 2 must be provided. Why were the values changed?
If the input values where change why the model results are remaining the same?
Author Response
Author's Reply to the Review Report (Reviewer 2)
General response
Authors sincerely thank the reviewer for his valuable comments in the second round, which were of great help in revising the manuscript.
Comments:
- The MQL oil specification must be also included for the cross-validation of the results.
Authors’ reply: The authors would like to thank the reviewer for his valuable comment. MQL oil specification has been added to the revised manuscript.
- The reason behind the change in the values on table 2 must be provided. Why were the values changed?
- If the input values where change why the model results are remaining the same?
Authors’ reply: Our sincere apologies for not clarifying the change made in table 5 in the revised submission (table 2 in initial submission). When reviewing the results database to implement the requested changes, it was noticed that were some typographical errors when writing the values in the table. Therefore the results are still the same while the input parameter values, which were wrongly stated due to the typo error, have been changed.
Round 3
Reviewer 2 Report
The authors have addressed the reviewers comments.